# Quantitative MRI reveals differences in striatal myelin in children with DLD

Saloni Krishnan[1,2]*, Gabriel J Cler[1,3], Harriet J Smith[1,4], Hanna E Willis[1,5], Salomi S Asaridou[1], Máiréad P Healy[1,6], Daniel Papp[7,8], Kate E Watkins[1]

[1]Wellcome Centre for Integrative Neuroimaging, Dept of Experimental Psychology, University of Oxford, Oxford, United Kingdom; [2]Department of Psychology, Royal Holloway, University of London, Egham Hill, London, United Kingdom; [3]Department of Speech and Hearing Sciences, University of Washington, Seattle, United States; [4]MRC Cognition and Brain Sciences Unit, University of Cambridge, Cambridge, United Kingdom; [5]Nuffield Department of Clinical Neurosciences, John Radcliffe Hospital, Oxford, United Kingdom; [6]Department of Psychology, University of Cambridge, Cambridge, United Kingdom; [7]NeuroPoly Lab, Biomedical Engineering Department, Polytechnique Montreal, Montreal, Canada; [8]Wellcome Centre for Integrative Neuroimaging, FMRIB Centre, Nuffield Department of Clinical Neuroscience, University of Oxford, Oxford, United Kingdom

**Abstract** Developmental language disorder (DLD) is a common neurodevelopmental disorder characterised by receptive or expressive language difficulties or both. While theoretical frameworks and empirical studies support the idea that there may be neural correlates of DLD in frontostriatal loops, findings are inconsistent across studies. Here, we use a novel semiquantitative imaging protocol – multi-parameter mapping (MPM) – to investigate microstructural neural differences in children with DLD. The MPM protocol allows us to reproducibly map specific indices of tissue microstructure. In 56 typically developing children and 33 children with DLD, we derived maps of (1) longitudinal relaxation rate R1 (1/T1), (2) transverse relaxation rate R2* (1/T2*), and (3) Magnetization Transfer saturation (MTsat). R1 and MTsat predominantly index myelin, while R2* is sensitive to iron content. Children with DLD showed reductions in MTsat values in the caudate nucleus bilaterally, as well as in the left ventral sensorimotor cortex and Heschl's gyrus. They also had globally lower R1 values. No group differences were noted in R2* maps. Differences in MTsat and R1 were coincident in the caudate nucleus bilaterally. These findings support our hypothesis of corticostriatal abnormalities in DLD and indicate abnormal levels of myelin in the dorsal striatum in children with DLD.

*For correspondence:
saloni.krishnan@rhul.ac.uk

Competing interest: The authors declare that no competing interests exist.

## Editor's evaluation

The work establishes changes in striatal and cortical myelin as a neural basis for developmental language disorder (DLD). This is a significant advance in the understanding of this disorder that will stimulate further work.

## Introduction

Children with developmental language disorder (DLD) struggle to learn their native language for no obvious reason. DLD is an extremely common neurodevelopmental disorder, with recent estimates indicating the prevalence of DLD is 7% (*Norbury et al., 2016*). DLD has serious economic and social consequences – it is associated with a higher risk for academic underachievement, unemployment, social and behavioural difficulties, and detriment to well-being (*Conti-Ramsden et al., 2018*).

**eLife digest** Seven percent of children struggle to learn their native language for no obvious reason. This condition is called Developmental Language Disorder (DLD). Children with DLD often have difficulty learning to read and write. They are at higher risk for academic underachievement and may struggle to find good jobs. Their language difficulties also contribute to difficulties making friends and emotional challenges.

Scientists suspect children with DLD may have differences in areas deep in the brain that help people learn habits and rules. A new magnetic resonance imaging technique called multiparameter mapping (MPM) can help scientists determine if this is true. The technique measures the properties of brain tissue. It is particularly useful for measuring the amounts of a fatty protective sheath on brain cells called myelin. Myelin helps brain cells send information faster.

Using MPM, Krishnan et al. show that children with DLD have less myelin in parts of the brain responsible for speaking, listening, and learning rules and habits. In the experiments, 56 children with typical language development and 33 children with DLD were scanned using MPM. Krishnan et al. then compared the two groups and found reduced myelin in these critical areas associated with learning a language in most of the children with DLD. But not all children with DLD had these differences.

More studies are needed to determine if these brain differences cause language problems and how or if experiencing language difficulties could cause these changes in the brain. Further research may help scientists find new treatments that target these brain differences.

Although we know that DLD does not result from gross neural lesions, we still do not have a clear picture of how brain anatomy differs in children with DLD (*Krishnan et al., 2016*; *Mayes et al., 2015*). This is not only practically relevant but would also help us to understand the neural underpinnings of language development. Here, we use a robust new semiquantitative imaging protocol – MPM or multi-parameter mapping (*Weiskopf et al., 2013*; *Weiskopf et al., 2021*) – to shed light on microstructural neural differences in a large group of children with DLD.

There is a dearth of literature examining brain structure in children with DLD, which is surprising given the prevalence and impact of DLD. In the available literature, grey matter changes have been noted in the left inferior frontal gyrus and the posterior superior temporal gyrus (*Badcock et al., 2012*; *Gauger et al., 1997*; *Jäncke et al., 2007*; *Lee et al., 2020*; *Plante, 1991*; *Preis et al., 1998*). These are core regions for language processing, with language activation in these regions observed across the literature (*Price, 2012*) and across languages (*Malik-Moraleda et al., 2021*). Differences have also been noted in areas homologous to these language regions, such as the right perisylvian cortex (*Girbau-Massana et al., 2014*; *Jäncke et al., 2007*; *Kurth et al., 2018*). However, there is variability in the direction of differences reported in different studies – for instance, both increases and decreases in grey matter have been noted in the left inferior frontal gyrus (*Badcock et al., 2012*; *Gauger et al., 1997*; reviewed in *Mayes et al., 2015*).

In addition to these cortical changes in the language network, we, and others, have hypothesised that the dorsal striatum is important for language learning, and may be abnormal in DLD (*Krishnan et al., 2016*; *Ullman et al., 2020*; *Ullman and Pierpont, 2005*). The dorsal striatum is important for habitual and sequential learning (*Graybiel and Grafton, 2015*; *Yin and Knowlton, 2006*), and we hypothesise that it may play an important role in the acquisition of language because of the complexity of sequencing required for language. In accordance, a series of behavioural studies has suggested that sequential learning in the linguistic and non-linguistic domains is affected in children with DLD (*Hsu and Bishop, 2014*; *Lum et al., 2014*; but see *West et al., 2018*; *West et al., 2021*). In studies that probe complex sequential production in the vocal domain, the dorsal striatum is implicated (*Rauschecker et al., 2008*; *Simmonds et al., 2014*; *Skipper et al., 2020*). The striatum is structurally and functionally connected to regions associated with language production, with the head of the dorsolateral caudate nucleus receiving inputs from inferior frontal cortex, and the putamen receiving inputs from motor, premotor, and supplementary motor cortex (*Alexander et al., 1986*; *Jarbo and Verstynen, 2015*; *Lima et al., 2016*). The importance of the dorsal striatum for speech and language was first highlighted by work on the KE family, who have a point mutation in the *FOXP2*

gene and a behavioural profile similar to DLD (**Watkins et al., 2002a**), as well as childhood apraxia of speech. Morphometric studies revealed that affected members of the KE family had reduced grey matter in regions not typically associated with language processing such as the head of the caudate nucleus, areas within the sensorimotor cortex, the posterior inferior temporal cortex and the posterior lobe of the cerebellum, and increased grey matter in the putamen (**Argyropoulos et al., 2019**; **Belton et al., 2003**; **Watkins et al., 2002b**). These studies indicate that corticostriatal circuits might have a role in co-ordinating and learning the fine auditory-motor sequencing required for language.

Studies of children with DLD analysing standard T1-weighted scans have also indicated reductions in the size of the caudate nucleus (**Badcock et al., 2012**; **Herbert et al., 2003**; **Jernigan et al., 1991**). Other studies do not wholly support the view that the volume of the caudate nuclei is reduced in DLD but do indicate that there is some abnormality. **Lee et al., 2013** found reduced absolute volumes in the caudate nucleus and thalamus in individuals with DLD. These relationships did not survive correcting for total intracranial volume, which was significantly reduced in their DLD participants. The authors also observed negative relationships between language proficiency and the relative volume of subcortical structures such as the nucleus accumbens, globus pallidus, putamen, and hippocampus. Others have suggested that volumetric differences in the caudate nuclei are modulated by age, with only younger children showing differences in volume (**Soriano-Mas et al., 2009**). Finally, some more recent studies suggest that children with DLD have greater grey matter in the right cerebellum (**Pigdon et al., 2019**). The interpretation of both cortical and subcortical findings is complicated by the heterogeneity of the DLD populations sampled, and the small sample sizes investigated. However, another factor that may also contribute to this lack of clarity is the nature of the scans acquired.

Standard structural imaging protocols such as T1-weighted scans reflect a complex mix of tissue properties, or in other words, the contrast between grey and white matter reflects a combination of histological properties such as iron content, myelin, cell density, and water. Importantly, these micro-structural properties yield regionally specific contributions to commonly used structural markers such as grey matter volume or cortical thickness, which complicate the interpretation of these markers (**Lorio et al., 2014**; **Lorio et al., 2016**). As standard T1-weighted imaging protocols are dependent on acquisition parameters that can vary across scanners, they are also often difficult to replicate across studies. More recently, semiquantitative MRI methods have been used to map-specific properties of tissue (**Weiskopf et al., 2021**). Semiquantitative protocols such as MPM can provide specific indices of microstructure, including myelin and macromolecular content of neural tissue, and the resulting maps are highly reproducible across individuals and scanners (**Leutritz et al., 2020**; **Weiskopf et al., 2013**). In the MPM quantitative imaging protocol, multiple maps are constructed, which allow us to probe different tissue properties. The generated maps quantify (1) the longitudinal relaxation rate R1 [1/T1], (2) the transverse relaxation rate R2* [1/T2*], and (3) Magnetization Transfer saturation (MTsat). The dominant influence on R1 in cortical tissue is myelin (**Lutti et al., 2014**), although R1 is sensitive to both myelin and iron in subcortical grey matter. R2* is sensitive to iron concentration, especially in ferritin-rich regions such as the striatum (**Langkammer et al., 2010**). MTsat is sensitive to bound water, and consequently myelin (**Schmierer et al., 2004**). This quantitative protocol therefore represents an unparalleled means of acquiring time-efficient, multi-modal, whole-brain data with insight into tissue composition. Such semiquantitative maps have been used to delineate heavily myelinated areas such as somatomotor (**Carey et al., 2017**), visual (**Sereno et al., 2013**), and auditory cortex (**Dick et al., 2012**). They have also been used to characterise developmental maturation in adolescence and young adulthood (**Carey et al., 2018**; **Ziegler et al., 2019**; **Paquola et al., 2019**; **Whitaker et al., 2016**), during ageing (**Callaghan et al., 2014**; **Draganski et al., 2011**; **Steiger et al., 2016**), and in pathological populations (**Freund et al., 2013**; **Manara et al., 2019**). A growing number of studies use these maps to understand brain–behaviour relationships (**Allen et al., 2017**; **Clark et al., 2020**). Most recently, we have used MPMs and found elevated iron levels in the putamen and speech motor network in people who stutter (**Cler et al., 2021**). However, this novel semiquantitative protocol has not yet been used to examine microstructure in children with neurodevelopmental disorders. A particular advantage of this protocol is that its sensitivity to cortical myelin can help distinguish two different explanations of developmental change in grey matter. Developmentally, cortical thinning indexed through grey/white matter contrast changes in standard T1-weighted scans could reflect a loss in the number of connections within grey matter, that is 'synaptic pruning' (**Huttenlocher, 1979**), or a gain in the volume of tissue through increased intra-cortical myelination that appears to 'whiten' the grey

**Table 1.** Descriptive data for the typically developing (TD), developmental language disorder (DLD), and history of speech and language (HSL) difficulties groups.

Means are shown below, with standard deviations in parentheses. Language proficiency and memory factor scores are derived from a factor analysis (for a full description, see *Krishnan et al., 2021*). Nonverbal IQ is a scaled score ($M = 10$, SD = 3) representing an average of performance on block design and matrix reasoning tasks. The last column shows whether there were significant group differences when using *t*-tests ($p < 0.05$), no correction for multiple comparisons is applied.

|  | TD | DLD | HSL | Group differences |
|---|---|---|---|---|
| Age (years) | 12.41 (1.62) | 12.48 (1.80) | 12.40 (1.67) | None |
| Gender | 28 F:28 M | 11 F:22 M | 3 F:17 M | N/A |
| Total intracranial volume (mm³) | 1329.24 (145.98) | 1345.39 (145.43) | 1411.51 (158.99) | None |
| Language proficiency | 0.8 (0.45) | −0.95 (0.55) | −0.06 (0.48) | TD > HSL > DLD |
| Memory | 0.58 (0.80) | −0.74 (0.79) | 0.07 (0.77) | TD > HSL > DLD |
| Nonverbal IQ | 12.3 (1.91) | 8.65 (2.09) | 11.0 (1.83) | TD > HSL > DLD |

matter on T1-weighted images (*Paus, 2005*). Recent studies using MPM protocols have shown that cortical thinning during development is associated with increased myelination rather than synaptic pruning (*Natu et al., 2019*; *Whitaker et al., 2016*).

In the present study, we used the MPM quantitative imaging protocol to map contrast parameters (R1, R2*, and MTsat) in typically developing (TD) children and those with DLD. Based on our previous studies (*Badcock et al., 2012*; *Watkins et al., 2002b*), we hypothesised that these indices would reveal that the microstructure of (1) the dorsal striatum (the caudate nuclei and the putamen) and (2) the left inferior frontal gyrus was altered in those with DLD.

## Results

As part of the Oxford BOLD study, we collected brain imaging data from 162 children between the ages of 10 and 15 years, as well as performing a detailed characterisation of their language and cognitive skills. All children in the study had a nonverbal IQ >70. Children were categorised as DLD if they scored 1SD below the mean on two or more language tests ($N = 57$), and TD ($N = 77$) if they scored ±1 SD of the mean on language tests (see Methods for more detail). Behavioural testing in a further 28 revealed that they did not meet our criteria for DLD but presented with a history of speech and language (HSL) problems. After quality control, we retained MPM data from 109 of these children,

**Table 2.** Group mean and standard deviation of parameter values in grey and white matter in children who were typically developing (TD), had developmental language disorder (DLD), and published mean and standard deviation in adults, mean 24.2 years, SD 1.6 years (*Weiskopf et al., 2013*).

Values in bold indicate a global difference between the TD and DLD groups ($p < 0.05$), uncorrected for multiple comparisons.

|  | TD | DLD | Published values |
|---|---|---|---|
| *Grey matter* | | | |
| MT | 0.828 (0.018) | 0.821 (0.019) | 0.794 (0.014) |
| R1 | ***0.615 (0.012)*** | ***0.608 (0.014)*** | 0.609 (0.008) |
| R2* | 15.192 (0.437) | 15.234 (0.544) | 15.200 (0.400) |
| *White matter* | | | |
| MT | 1.720 (0.052) | 1.714 (0.053) | 1.764 (0.066) |
| R1 | 0.973 (0.025) | 0.962 (0.025) | 1.036 (0.036) |
| R2* | 20.672 (0.659) | 20.679 (0.704) | 21.000 (0.800) |

including 56 TD children, 33 children with DLD, and 20 children with HSL. The children in the HSL group were excluded from comparisons of TD and DLD, but were included in continuous analyses which allow us to examine language variability. The three groups (TD, HSL, and DLD) did not differ in terms of age (see *Table 1*).

## Whole-brain comparisons of neural microstructure in children with DLD with TD children

We first compared children who met criteria for DLD and TD children. There were no group differences in mean MTsat and R2* values for grey and white matter, or in total intracranial volume. The groups did differ in mean R1 values for grey matter (see *Table 2*), in that children with DLD had lower R1 than the TD group across all grey matter. We created averages of MTsat, R1, and R2* for each group. Across all three maps, we observed high values in primary motor, visual, and auditory cortex (see *Figure 3—figure supplement 1*), in line with our expectations. Additionally, we observed a close correspondence between average values of MTsat, R1, and R2* in grey and white matter in our two groups and published values (*Weiskopf et al., 2013*), see *Table 2*.

We then investigated whether there were group differences in neural microstructure by assessing each of the parameter maps using nonparametric permutation methods and established significant clusters using the threshold-free cluster enhancement method, setting a whole-brain corrected threshold of *p*<.05. Children with DLD had lower MTsat values than TD children in the inferior frontal gyrus (pars opercularis), ventral sensorimotor cortex, insular cortex, lateral Heschl's gyrus, planum temporale, and posterior superior temporal sulcus of the left hemisphere, and in portions of lateral and dorsomedial occipital cortex bilaterally. Subcortically, the children with DLD also had reduced MTsat in the dorsal caudate nucleus bilaterally; these differences were seen mainly in the body and were more extensive on the left than the right (*Figure 1* and *Supplementary file 1a*). Consistent with the mean global differences in R1 in children with DLD (see *Table 2*), examination of the R1 maps revealed widespread reduction over the lateral convexities of the frontal and parietal lobes bilaterally (but slightly more on the right), the medial frontal cortex including SMA and extending to paracentral lobule and left posterior temporal cortex extending from the superior temporal plane to posterior inferior temporal cortex. Subcortically, there were differences in R1 in the dorsal striatum and thalamus bilaterally, and in anterior portions of the medial temporal lobe (*Figure 2*). We did not find any significant group differences when examining the R2* maps. There were also no regions where children with DLD showed greater MT, R1, or R2* values relative to TD children.

Given that MTsat and R1 maps are particularly sensitive to myelin, and the R2* maps did not differ between the groups, we then examined if there were regions where we would see convergence of differences across MTsat and R1. A conjunction analysis was performed, where we assessed which voxels showed significant TD > DLD differences in both MTsat and R1 maps. We found conjoint differences in the MT and R1 maps across several brain regions, including the caudate nuclei bilaterally, and in the left ventral sensorimotor cortex, insula, lateral Heschl's gyrus, planum temporale, posterior superior temporal sulcus, and middle temporal gyrus (*Figure 3* and *Supplementary file 1b*).

The CATALISE definition of DLD (*Bishop et al., 2016*; *Bishop et al., 2017*) rightly removes the criterion requiring a discrepancy between verbal and nonverbal skills, which in practice means broadening the phenotype to include children with low nonverbal IQ. To determine whether our group differences were in fact driven by the inclusion of children with low nonverbal IQs in addition to low language ability, we assessed differences only in children with nonverbal IQ scores >85. Group differences in MTsat and R1 maps were observed even when removing the five children with DLD who had nonverbal IQs between 70 and 85 (see *Figure 1—figure supplement 1* and *Figure 2—figure supplement 1*), indicating the differences reported here were not driven by the inclusion of children with low nonverbal IQs. Consequently, all following analyses include these children.

Given our interest in the dorsal striatum in relation to DLD (see Introduction), we extracted average MTsat values for each participant from the caudate nuclei bilaterally (the region where we observed TD > DLD differences). Using a hemispheric mask, we separated this cluster into left and right components. We then assessed if group differences in average MTsat values in the left and right caudate nuclei could be accounted for by age or total intracranial volume. Age and total intracranial volume were not significant predictors of MTsat values in the caudate nuclei clusters, see *Figure 1—figure supplement 2*.

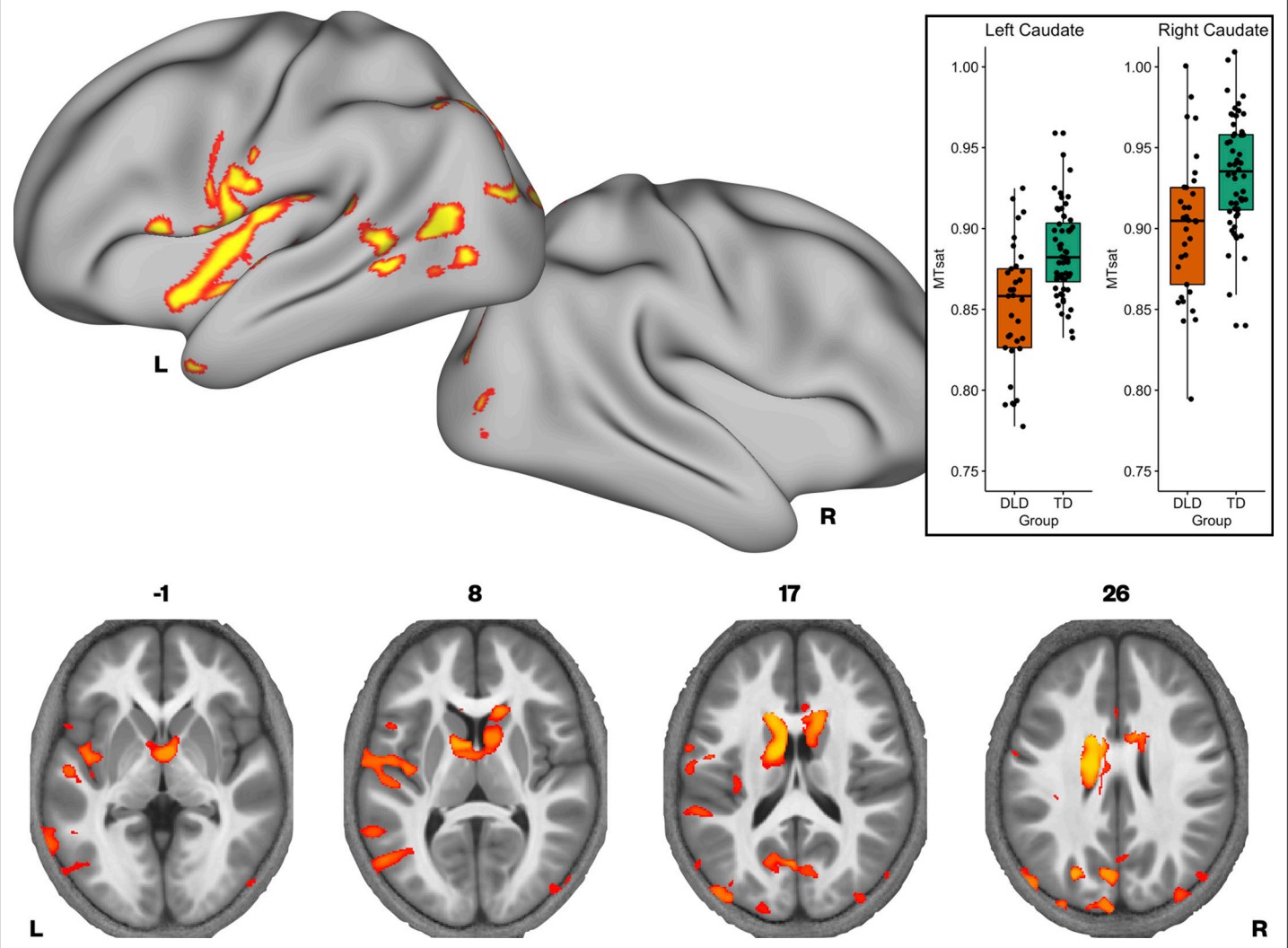

**Figure 1.** Brain areas showing reduced Magnetization Transfer saturation (MTsat) in developmental language disorder (DLD). Coloured maps resampled and overlaid on the fsaverage surface (reconstructed using the HCP workbench) show TD > DLD differences in MTsat values (whole-brain corrected threshold *p* < 0.05). Axial slices (coloured maps are overlaid on an average MTsat image from all participants) show additional group differences subcortically in the caudate nuclei. The inset shows a boxplot of MTsat values drawn from the cluster in the caudate nuclei by group, with this cluster split into right and left using a hemispheric mask (orange – DLD, green – TD).

The online version of this article includes the following figure supplement(s) for figure 1:

**Figure supplement 1.** Whole-brain typically developing (TD) > developmental language disorder (DLD) differences (thresholded at p < 0.05) when excluding children with nonverbal IQs < 85 in Magnetization Transfer saturation (MTsat) values.

**Figure supplement 2.** Relationship between MTsat values in the caudate nucleus by age and total intracranial volume (TIV) in typically developing (TD) children and those with developmental language disorder (DLD).

## Whole-brain correlation analysis of neural microstructure with language proficiency

We have previously found continuous measures of language proficiency to be more sensitive to neural differences than diagnostic categories (*Krishnan et al., 2021*). Using a continuous measure of language ability also gave us the opportunity to include the group of children with HSL problems who did not meet criteria for DLD on testing (HSL; *N* = 20). We constructed language and memory factor scores from our neuropsychological battery (see *Krishnan et al., 2021* for further detail on model construction), using data from the full sample in whom behavioural data were available in the BOLD study. We then assessed if these factors, as well as nonverbal IQ, were predictors of our three parameter values across our whole sample (*N* = 109) using whole-brain analyses. Given the strong correlation between

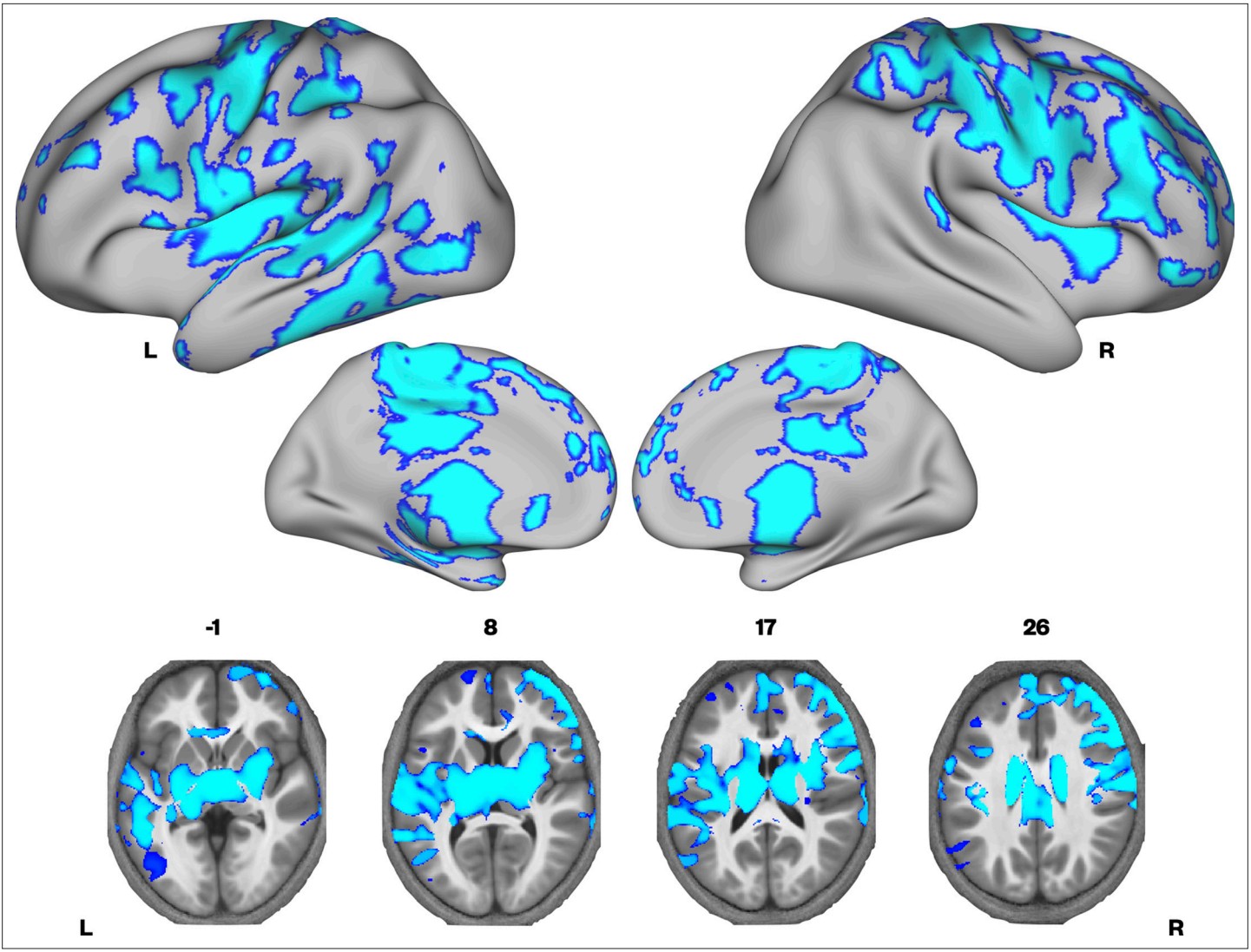

**Figure 2.** Brain areas showing reduced R1 in developmental language disorder (DLD). Coloured maps resampled and overlaid on the fsaverage surface (reconstructed using the HCP workbench) show TD > DLD differences in R1 maps (whole-brain corrected threshold $p < 0.05$). Axial slices (coloured maps are overlaid on an average MTsat image from all participants) show additional subcortical group differences in the striatum and thalamus bilaterally.

The online version of this article includes the following figure supplement(s) for figure 2:

**Figure supplement 1.** Whole-brain typically developing (TD) > developmental language disorder (DLD) differences (thresholded at p < 0.05) when excluding children with nonverbal IQs <85 in R1 values.

the language and memory factors ($r = 0.7$, p < 0.001), we entered these predictors separately into our statistical models. We found that language proficiency was strongly positively correlated with MTsat values focally in the left caudate nucleus (*Figure 4*). In the R1 maps, poorer language proficiency was once again associated with widespread reduction over the lateral convexities of the frontal and parietal lobes bilaterally (as seen before in the TD > DLD differences, which were slightly more right lateralised), the medial frontal cortex including SMA and extending to paracentral lobule, and left posterior temporal cortex extending from the superior temporal plane to posterior inferior temporal cortex (see *Figure 4—figure supplement 1*). Subcortically, language proficiency was positively associated with R1 values in the dorsal striatum and thalamus bilaterally, and in anterior portions of the medial temporal lobe. This pattern of results was similar to those derived in the TD > DLD contrast. Lower R1 values in a more focal but overlapping set of regions (perisylvian cortex including the inferior frontal gyrus, insula, superior temporal gyrus, extending to the anterior temporal pole, bilaterally, but more extensive on the left, and the dorsal striatum bilaterally) were associated with poor memory

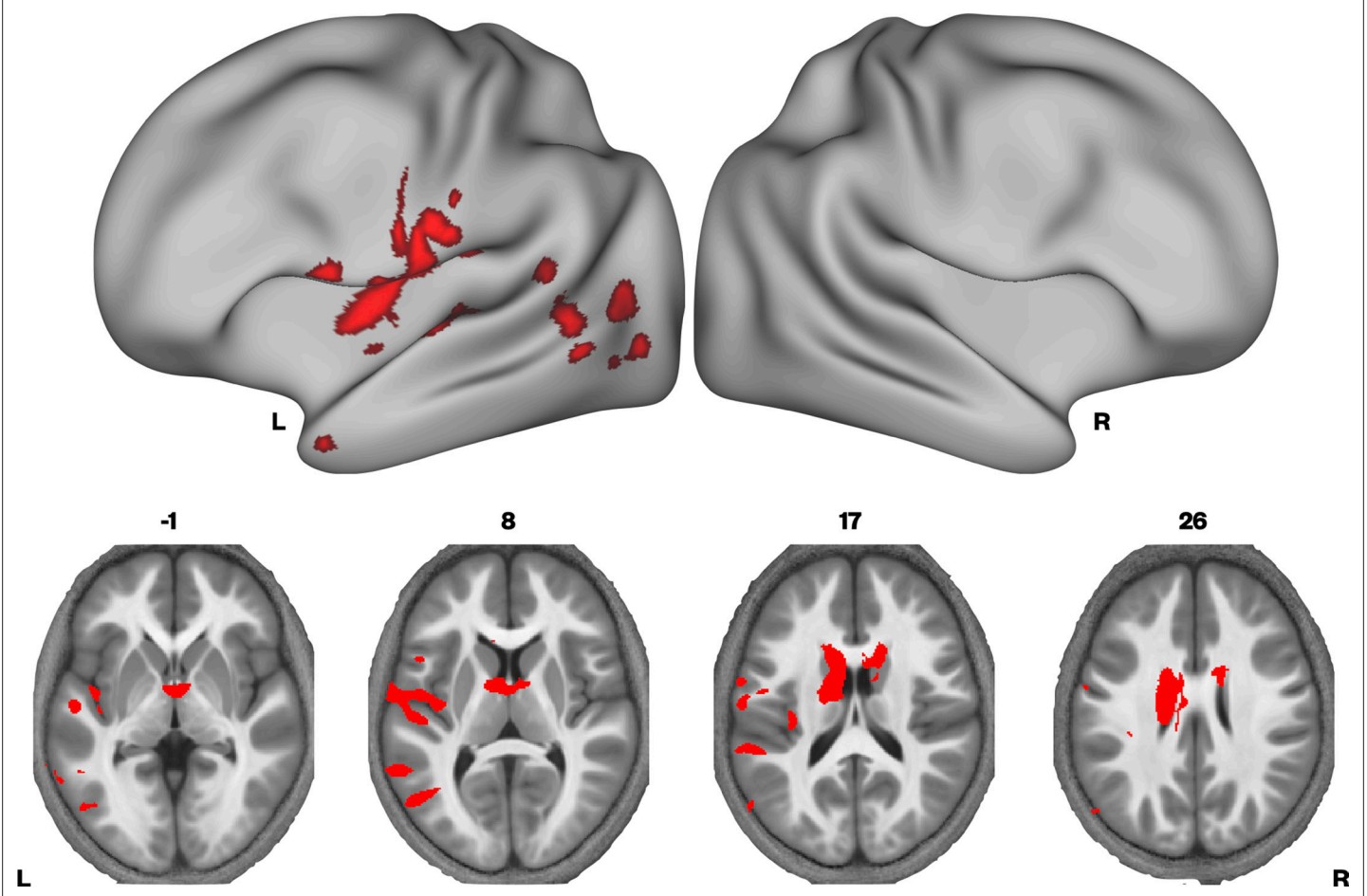

**Figure 3.** Brain areas showing conjoint reductions in MTsat and R1 in developmental language disorder (DLD). Coloured maps resampled and overlaid on the fsaverage surface (reconstructed using the HCP workbench) show convergence of TD > DLD differences in MTsat and R1 maps thresholded at p < 0.05. Axial slices (coloured maps are overlaid on an average MTsat image from all participants) show the differences subcortically in the dorsal striatum in particular.

The online version of this article includes the following figure supplement(s) for figure 3:

**Figure supplement 1.** Average MTsat, R1, and R2* values in typically developing (TD) and developmental language disorder (DLD) groups are shown using a coloured heat map (hotter colours show higher values).

proficiency, suggesting that these cognitive differences were reflecting the globally lowered values of R1 in children with DLD. There were no significant relationships between R2* values in grey matter and language or memory proficiency.

## Associations with language proficiency, memory, and IQ in the dorsal striatum

The DLD and TD groups differed on language proficiency, memory, and nonverbal IQ (see *Table 1*). We consequently assessed whether variation in language proficiency, memory, or nonverbal IQ best explained the variation in R1 and MTsat values in the portions of the caudate nucleus where we observed TD > DLD group differences. We constructed stepwise regressions to evaluate the contribution of each of these factors, including children from the HSL group to maximise power (the pattern of analysis was the same when we limited our analyses to children in the TD and DLD groups).

For MTsat values in the caudate nucleus, we found that a model with language proficiency alone ($\beta$ = 0.014, p < 0.001) was the best fitting model, explaining 14.57% of the variance, $F(1,106)$ = 18.08, p < 0.001. For R1 values in the caudate nucleus, we again observed that a model with language

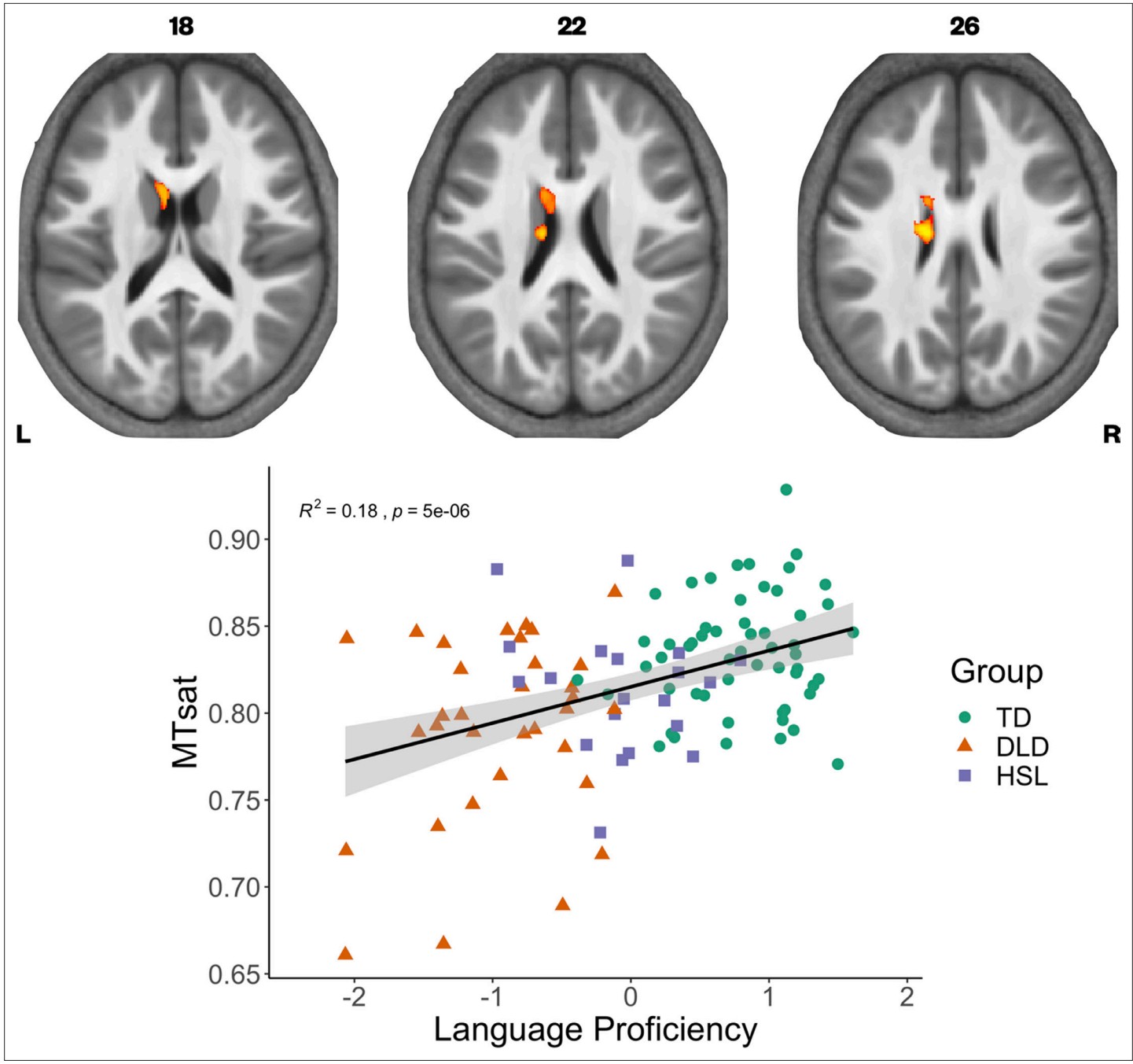

**Figure 4.** Correlation between language proficiency and Magnetization Transfer saturation (MTsat) values in the left caudate nucleus. Coloured maps are overlaid on axial slices through the average MTsat image from all participants (top) showing significant relationship subcortically in the left caudate nucleus. Average MTsat values for the left caudate nucleus in individual participants are plotted against the Language Proficiency Factor score. The solid line is the regression line with shaded areas showing the 95% confidence interval. Groups are plotted in different colours (green - TD, orange - DLD. purple - HSL) for illustration.

The online version of this article includes the following figure supplement(s) for figure 4:

**Figure supplement 1.** Whole-brain correlation of multi-parameter mapping (MPM) values with language proficiency (thresholded at p < 0.05).

proficiency alone ($\beta$ = 0.015, p < 0.001) was the best fitting model, explaining 13.19% of the variance, $F$(1,106) = 16.11, p < 0.001.

## TD vs. DLD group differences in voxel-based morphometry

Finally, to assess if the parametric differences reported here were also observed in standard morphometric analysis, we performed a voxel-based morphometry (VBM) analysis comparing regional amounts of grey matter in TD vs. DLD using T1 scans from the same participants (note these were collected using an MPRAGE T1-weighted sequence and were not the T1-weighted scan from the MPM protocol). We did not observe any group differences that survived our statistical threshold (p < 0.05), and indeed no group differences were observed at a lower statistical threshold p < 0.2. Thus, our parametric differences cannot be explained by morphometric differences in the amount of grey matter.

# Discussion

In this study, we used semiquantitative structural MRI to provide a detailed account of the neural differences in children with DLD, taking advantage of the sensitivity of this protocol to understand changes in neural microstructure. We found reduced MTsat and R1 values in the caudate nuclei in children with DLD. We also observed reduced MTsat values in the left inferior frontal gyrus. This offers empirical confirmation of our prediction that cortico-striato-thalamo-cortical loops involved in learning are affected in DLD.

Importantly, these results allow us to understand the cellular mechanisms driving this change. MTsat and R1 are considered in vivo markers of macromolecular content, and within the brain, these measures are particularly sensitive to myelin content in grey matter. Although most of the brain's myelin is found in the white matter where it sheaths the long axons travelling in white matter fibre tracts, it can also be measured in grey matter where it myelinates axons or parts of axons in the cortex and in subcortical structures like the thalamus and basal ganglia (*Nieuwenhuys, 2013*). Indeed, myelin is a strong contributor to MR signal in these regions (*Glasser and Van Essen, 2011*; *Sereno et al., 2013*; *Shafee et al., 2015*). Given myelin's role in enabling fast and reliable communication in the brain, reduced myelin content may explain why children with DLD struggle with speech and language processing. Below, we discuss these findings, contextualising why myelin may be altered in these specific regions in DLD.

## Interpretation of MPM scans

Myelin is known to increase throughout the brain during adolescence (*Paquola et al., 2019*; *Whitaker et al., 2016*), and has been linked to improved behavioural performance (*Kwon et al., 2020*). Previous morphometric studies have typically drawn inferences about myelin from cortical thinning, but shrinkage of grey matter does not allow us to distinguish if T1 change is due to shorter T1 times associated with reduced synaptic density, or an increase in the proportion of myelinated neurons (*Paus, 2005*). More recently, MTsat and R1 measures have been used as an in vivo marker for myelin. These have been validated as markers of myelin through postmortem imaging, as well as histological studies of patients with multiple sclerosis (*Weiskopf et al., 2021*). Importantly, these maps agree with histological maps showing greater myelin in primary motor and sensory cortex (*Carey et al., 2017*; *Dick et al., 2012*; *Sereno et al., 2013*; *Whitaker et al., 2016*), as can be seen in our maps as well (*Figure 3—figure supplement 1*). It is therefore unsurprising to find some convergence over MTsat and R1. R2* is also considered a measure of myelin, but the T2* contrast is particularly sensitive to iron, especially within the basal ganglia. Despite R2* differences being noted in adults with speech problems in the putamen and cortical speech motor network (developmental stuttering, *Cler et al., 2021*), and observed links between verbal memory and R2* in the ventral striatum (*Steiger et al., 2016*), we did not see any evidence for R2* differences anywhere in the brain when comparing our TD and DLD groups.

Our findings also strongly suggest that there is an advantage to using MPM to probe microstructure relative to standard T1 measures. While previous studies reported differences using morphometric measures, we did not observe evidence for morphometric differences in our sample. As others have argued, MPM is sensitive to microstructural differences that would not be detected by a standard

voxel-based morphometry (VBM) analysis, as VBM is sensitive to differences in regional amounts of grey matter based on T1 weighting rather than quantitative measurements (*Lorio et al., 2014*; *Lorio et al., 2016*). Using MPMs allows us to more closely interrogate the histological processes involved in neural changes.

## Group differences in the striatum

Our findings clearly indicate that there are microstructural abnormalities in the caudate nuclei bilaterally in children with DLD, and more broadly, that reduced myelin in the left caudate nucleus is associated with lower language proficiency. The convergence across MT and R1 differences in the caudate nuclei strongly suggest these are linked to abnormal levels of myelin.

The dorsal striatum has been implicated in learning through practice, particularly habit formation (*Skipper et al., 2020*; *Yin and Knowlton, 2006*). A number of groups, including us, have hypothesised that the striatum is a brain region where we might expect to see neural differences in children with DLD (*Krishnan et al., 2016*; *Ullman et al., 2020*; *Ullman and Pierpont, 2005*). Our hypothesis was driven by the idea that children with DLD showed deficits in sequential procedural tasks, which rely on loops through the striatum (*Krishnan et al., 2016*). Here, using a direct neural measure, we see evidence for structural differences in the striatum, particularly in the caudate nuclei. The functional consequences of these structural changes might be to make certain aspects of learning, such as the learning of stimulus-response mappings, more challenging. For instance, a reduction in myelin might make sequential learning less efficient. This could explain why children with DLD face difficulties in complex language tasks, such as nonword repetition, where extracting and producing sequential responses are important. However, any speculation about the functional impact of these changes needs careful empirical investigation in future studies.

Our findings are generally consistent with previous smaller-scale work linking volumetric differences in the caudate nucleus to language learning difficulties (*Badcock et al., 2012*; *Watkins et al., 2002b*). They may also offer some resolution to previous work using VBM, where such differences were not observed (*Pigdon et al., 2019*). Indeed, our own VBM analysis did not reveal any differences in morphometry. However, and perhaps speaking to a slightly different interpretation of our findings, the striatal differences we find are in the body of the caudate nucleus, rather than the head. The head of the caudate nucleus receives input from prefrontal cortex. DTI studies suggest the body of the caudate nucleus gets its input from prefrontal cortex and supplementary motor regions (*Lehéricy et al., 2004*). Functional connectivity studies indicate that the body of the caudate nucleus also receives projections from temporal association cortex (*Choi et al., 2012*). Given our findings of differences in the body rather than the head of the caudate nuclei, further studies examining individual differences of microstructure and relevant behavioural tasks (e.g. auditory processing and learning, or processing of rhythm) are warranted.

Although some previous work suggested that striatal differences were normalised by adolescence in children with DLD (*Soriano-Mas et al., 2009*), our analyses do not show any evidence of group differences being modulated by age (*Figure 1—figure supplement 2*). Using the MPM protocols may give us a more sensitive marker of change to differences in DLD. In these analyses, it is perhaps surprising that we did not see age-related change in the striatum within the time frame we sampled. Longitudinal work using MTsat has demonstrated that the striatum continues to mature through adolescence, from 14 to 24 years of age (*Paquola et al., 2019*). However, it may be that the time frame we sampled was too short, or that within-subject longitudinal studies, which are more sensitive to changes over time, are needed.

## Genetic mechanisms underlying group differences in the striatum

The genetic mechanisms that might drive these striatal changes are currently unclear. DLD is a highly heritable disorder but identification of genetic pathways has been difficult due to the broad phenotype and changes in diagnostic criteria (for a recent review, see *Mountford et al., 2022*). In terms of monogenic disorders, the best known case is that of the KE family, the affected members of which had a rare point mutation in the gene *FOXP2* causing verbal dyspraxia as well as the kinds of language learning difficulties seen in DLD. The mutation in *FOXP2* was associated with reduced volumes of the caudate nuclei bilaterally in the KE family (*Watkins et al., 2002b*), but it is unknown whether this was related to a quantitative difference in myelin content as seen here in DLD. *FOXP2* is strongly

expressed in the medium-spiny neurons of the striatum in humans and other species, and in many other neural and non-neural tissues (*Lai et al., 2003*). *FOXP2* is a transcription factor affecting the function of many downstream target genes, including *CNTNAP2* and *FOXP1*, which in turn have been linked to phenotypic features of DLD (*Lozano et al., 2015*; *Vernes et al., 2008*). Aside from identifying rare monogenic causes of DLD, other approaches have yielded common risk variants that explain variances in phenotypic features of DLD, including phonological skills (*Newbury et al., 2009*). A recent meta-analysis of DLD genome-wide association studies (GWAS) of reading and language traits with an *N* > 30,000 found significant genome-wide association with word reading and distinct genetic variation of word and nonword reading, spelling, and phoneme awareness that correlated with cortical surface area of the left superior temporal sulcus (*Eising et al., 2021*). Interestingly, in our study, the posterior part of the superior temporal sulcus in the left hemisphere showed significantly reduced MTsat and R1 values in children with DLD (see *Figures 1–3*).

## Neural differences beyond the striatum

We also observed MTsat changes indicative of myelin-related differences in the posterior part of the inferior frontal gyrus, ventral sensorimotor cortex, insula, planum temporale, lateral Heschl's gyrus, and the superior temporal sulcus, all in the left hemisphere. Regions such as the inferior frontal cortex and posterior superior temporal cortex are considered core parts of the speech and language network (*Malik-Moraleda et al., 2021*; *Fedorenko et al., 2011*; *Hickok and Poeppel, 2007*; *Rauschecker and Scott, 2009*), and ventral sensorimotor cortex and insula are regions that are clearly implicated in speech motor control (*Bouchard et al., 2013*; *Carey et al., 2017*; *Dronkers, 1996*; *Wise et al., 1999*). Our findings here of abnormal microstructure in these regions are somewhat consistent with those from previous studies in DLD that report structural differences in perisylvian regions (*Badcock et al., 2012*; *Gauger et al., 1997*; *Jäncke et al., 2007*; *Plante, 1991*; *Preis et al., 1998*) with two notable differences – one, that these differences did not emerge in a VBM analysis, and two, on average, children with DLD had lower MTsat or R1 values than TD children, indicative of slower maturation or abnormal developmental trajectories. Such differences in these regions therefore may be correlates of either auditory or motor inefficiency or both that have been observed in some children with DLD (*Halliday et al., 2017*; *Hill, 2001*; *McArthur and Bishop, 2004*).

While the TD > DLD differences observed in the MTsat appeared left lateralised and focal, differences in the R1 map were widespread, observed in both hemispheres quite symmetrically. Interpreting the differences in these findings offers a paradox. On the face of it, left-lateralised myelin reduction in brain regions known to contribute to speech and language processing seems very plausible, as this would be entirely consistent with the behavioural profile of DLD. This would fit with a popular theoretical view, that is that the left hemisphere is uniquely privileged to support language (*Vargha-Khadem et al., 1985*). However, this does not fit with the developmental literature on children with early brain lesions. Children with perinatal focal brain lesions, even those encompassing the entire left hemisphere, have fairly good language skills, and they typically perform better than children with DLD on language tasks (*Asaridou et al., 2020*; *Bates and Dick, 2002*; *Thal et al., 1991*). Right hemisphere homologues of language areas are able to support language reorganisation when early damage is sustained (*Bates and Dick, 2002*; *Newport et al., 2017*). This has led to theoretical speculation that abnormalities affect both hemispheres in children with DLD preventing this form of brain plasticity. In this vein, the widespread differences seen in the R1 maps, or the bilateral abnormalities observed in the caudate nuclei, might point to why organisation of language is not maximally efficient. Another possibility is that MTsat mainly indexes myelin differences, whereas R1 could be sensitive to other microstructural features such as iron and neuronal fibres (*Edwards et al., 2018*). This might suggest that the myelin differences observed in R1, that are not observed in MTsat, indicate further sources of neural difference in DLD. Further work is needed to understand what the divergent TD > DLD differences across the R1 and MTsat maps might reflect.

It is also notable that many of the MTsat and R1 differences we observe are in primary motor and sensory areas, or closely adjacent areas. Myelin content is high in primary sensory (due to dense thalamo-cortical projections) and motor areas (due to the large axons of cortico-spinal projections) and therefore peaks at earlier stages of development (*Natu et al., 2019*; *Paquola et al., 2019*; *Whitaker et al., 2016*). As seen from our average parameter maps (*Figure 3—figure supplement 1*), we see the expected strong myelination in these regions, and we may therefore have stronger signal

in these regions of the brain to evaluate group differences. In contrast, a relative lack of differences in association areas may be a true finding or may reflect reduced sensitivity to measure this change because there is less myelin content there. For instance, in our maps, we do not see very strong myelination in some cortical areas, such as the inferior frontal gyrus, which might limit the ability to see group differences. In TD teenagers, myelogenesis is highest in association areas (*Whitaker et al., 2016*). Longitudinal studies are therefore necessary to evaluate whether differences in myelin persist in the same areas in children with DLD, that is, regionally specific changes, or whether differences in myelin would be seen in association areas at later stages of development.

The broad constellation of challenges faced by children with DLD might also explain some of the group differences in MTsat and R1 values. For example, many children with DLD have co-occurring motor challenges (*Hill, 2001*; *Sanjeevan and Mainela-Arnold, 2019*), which may be reflected in changes in the motor network. This might also account for differences in regions we did not have clear hypotheses about, for example, in the occipital lobe. Many children in our study faced reading challenges, and left occipital–temporal hypoactivation is associated with dyslexia (*Paulesu et al., 2014*; *Richlan, 2012*) and now genetic variation in literacy skills (see above, *Eising et al., 2021*). One way to disentangle the contributions of these different cognitive traits would be to conduct large-scale studies including those with multiple diagnoses and use a transdiagnostic approach to establish specific relationships between brain and behaviour (*Siugzdaite et al., 2020*).

Finally, it is important to note that we previously hypothesised that the microstructure of grey and white matter in the medial temporal lobe would be relatively normal in children with DLD (*Krishnan et al., 2016*). However, we observed differences in R1 values in these regions (see *Figure 2*, medial surface of the left hemisphere). This fits with the emerging picture that children with DLD can struggle with aspects of learning thought to depend on the medial temporal lobe, for example declarative memory tasks such as list learning (*Bishop and Hsu, 2015*; *Earle and Ullman, 2021*; *Jackson et al., 2020*; *McGregor et al., 2017*).

## Implications for theories of DLD

The striatal differences we report are broadly consistent with our views on the dorsal striatum being involved in language learning (*Krishnan et al., 2016*), as well as with theories such as the procedural deficit hypothesis (*Lum et al., 2014*; *Ullman and Pierpont, 2005*), and the procedural deficit circuit hypothesis (*Ullman et al., 2020*). This may be linked to difficulties extracting and retaining the sequential regularities for language, or challenges in automatising the use of language rules. Our sample with DLD shows differences in linguistic tasks that draw upon these abilities, such as nonword repetition or oromotor sequencing, even though they were not selected on this basis. *Orpella et al., 2021* recently demonstrated that the dorsal and ventral striatum were crucial for developing and acting upon predictions in a statistical learning task. The use of such sensitive functional tasks may be important to demonstrate the functional consequences of these changes. Understanding such learning mechanisms may also help us understand how to design better intervention for children with DLD, for example, this may be a reason to simplify and repeat rules during learning, providing greater opportunities for practice and habituation. We are now testing the role of repetition in enhancing language comprehension in DLD (*Parker et al., 2021*).

Our more global differences in myelination would also be consistent with theoretical views postulating that children with DLD show differences in the speed of processing, perhaps due to inefficiencies in information transfer through the brain (*Kail, 1994*; *Miller et al., 2001*). However, more recent theories point to more specific lexical processing differences in DLD. Children with DLD have been shown to have greater difficulty with lexical selection and inhibition of competitors, rather than the initial perceptual or phonological processing (*Nation, 2014*). Computational models suggest this could be due to increased levels of lexical decay or lexical inhibition (*McMurray et al., 2010*; *Apfelbaum et al., 2022*). Abnormalities in myelination could result in differing levels of inhibition for targets and competitors. Techniques like MEG would be better suited to test this hypothesis.

There has also been significant theoretical debate about whether DLD represents the tail end of a spectrum of language ability or if it represents a group or several subgroups that are biologically distinguishable from those with typical language ability (*Tomblin, 2011*). Broadly speaking, there is now consensus in the field that these children represent the extreme end of a distribution (*Bishop et al., 2016*; *Lancaster and Camarata, 2019*; *Tomblin and Zhang, 1999*). We examined if our data

were consistent with this idea by examining the distribution. When plotted, the average values of striatal myelination for individual participants indicated overlap between the DLD and TD groups (see *Figure 1*). So not all children with DLD had values outside of the range of values for the TD groups, as might be expected if we were observing a distinct subtype. However, our continuous analysis does indicate that those who have the lowest values for striatal myelination have the lowest language factor scores (see *Figure 4*). Our limited exploration of how these measures are distributed both within the DLD group and across the cohort is consistent with the idea that DLD is a spectrum disorder. Even so, further theoretically driven analyses that relate specific behaviour to these quantitative measures of brain microstructure might provide insight into whether there are distinct subtypes within DLD. Future studies should focus on understanding the full spectrum of DLD using large datasets, establishing if there are distinct brain–behaviour relationships within this broad category.

### Limitations and future directions

The differences we see here are observed at a group level. In other words, as noted above, lower MTsat or R1 values in the caudate nuclei are not observed in every child with DLD. In future studies, we are keen to use structural connectivity analyses, as they will allow us to understand how differences across a network of brain areas may make children susceptible to DLD. It is also unclear whether the neural changes we observe are the cause of DLD, or a consequence of having a language disorder. Longitudinal studies where children are followed over time are the best way to shed light on this issue. A pertinent issue when considering longitudinal studies is the amount of data we were able to retain from children we tested (between approximately 65% and 80% based on whether children were in the TD, HSL, or DLD group, see Methods). We were concerned that systematic biases might affect retention, with children with severe language problems being more likely to be excluded. Consequently, in our sample, we assessed if there were any differences between children whose scans were excluded, relative to those we retained for our analysis. We found that children with DLD whose scans were excluded were younger than those who were selected, but they were not more severely affected in terms of their language learning (*Supplementary file 1c*). However, it is worth noting our analysis does not account for the children who we recruited but were unable to scan. Dropout and data quality are factors to consider if scanning younger children with this protocol. Finally, the relationship between structure and function is complex. We need to understand how these structural differences might affect specific aspects of function. For instance, we did not observe differences in dorsal striatal activity for a simple language task of verb generation in the same group of children (*Krishnan et al., 2021*). This may signal the task was not sensitive to differences, or alternately, that we need to tap different aspects of language processing.

### Summary and conclusions

Understanding the neural basis of DLD is particularly challenging given the developmental nature of the disorder, as well as the lack of animal models for understanding language. Novel semiquantitative MPM protocols allow us an unparalleled in vivo method to investigate microstructural neural changes in these children. Our findings using this protocol suggest that the caudate nucleus, as well as regions in the wider speech and language network, show alterations in myelin in children with DLD. These findings strongly point to a role for the striatum in the development of DLD. This role is likely to be in the learning of habits and sequences, but future work is necessary to test this hypothesis given the anatomical localisation in our study. Additionally, myelin patterns can change over development, and myelination can be observed after successful training. In next steps, it is important to assess whether these differences in myelin persist over development in DLD, and if they can be targeted through training using behavioural interventions.

## Materials and methods

### Participants

As part of the Oxford BOLD study, we recruited and tested 175 children between the ages of 10 and 15 years. All children had to meet certain inclusion/exclusion criteria; specifically, they had to have: (1) normal hearing (defined as passing audiometric screening at 25 dB at 500, 1000, and 2000 Hz, in the better ear); (2) a nonverbal IQ >70 (assessed using the WISC-IV Matrix Reasoning and Block Design

Tests – *Wechsler, 2004*); and (3) have grown up in the UK speaking English. Children were excluded if they had another neurodevelopmental disorder such as autism or attention-deficit hyperactivity disorder, or history of neurological disorder. Participants who met inclusionary/exclusionary criteria were categorised as having DLD if they presented with a history of language problems and scored at least 1 SD below the normative mean on two or more standardised tests of language ability. Children were categorised as HSL if they presented with HSL problems but did not meet criteria for DLD. Those who were categorised as TD had no history of speech and language problems. If these children scored 1 SD or more below the mean on more than one standardised test score of language ability, they were excluded from the TD group. Of the 175 children we recruited, a total of 162 children completed both behavioural testing and MRI scans and met our inclusionary criteria (77 TD children, 57 children with DLD, and 28 who had a history of speech and language (HSL) difficulties but did not meet our criteria for DLD at time of testing), for further details, see *Krishnan et al., 2021*. From this sample, we acquired MPM data in 72 TD children, 51 children with DLD, and in 25 children with HSL.

## Data acquisition

MR data were collected with a 3T Siemens Prisma scanner with a 32-channel head coil. Participants wore noise-cancelling headphones (Optoacoustics OptoActive II Active Noise Cancelling Headphones). Foam padding was placed around the head for comfort and to restrict movement; the headphones were held in place with inflatable pads.

Whole-brain images at an isotropic resolution of 1 mm were obtained using an MPM quantitative imaging protocol (*Lutti et al., 2014*; *Weiskopf et al., 2013*). This protocol consisted of the acquisition of three multi-echo gradient acquisitions with proton density (PD), T1, or MT weighting. Each acquisition had a TR of 25 ms, field of view = 256 × 224 × 176 mm$^3$, readout bandwidth 488 Hz/pixel, and slab rotation of 30°. Flip angle for MT- and PD-weighted acquisitions was 6°, and 21° for T1-weighted acquisitions. MT weighting was achieved by using a Gaussian radiofrequency (RF) pulse 2 kHz off resonance with 4-ms duration and a nominal flip angle of 220° prior to excitation. To speed up data acquisition, a GRAPPA acceleration factor of 2 was applied, with 40 references lines in each phase encoding direction. Eight echoes were acquired for the T1- and PD-weighted contrasts, and six echoes were acquired for the MT contrast. Each sequence took approximately 5 min to acquire. In addition, data to calculate an RF transmit field map was acquired at the start of the session, using a 3D echo-planar imaging spin-echo/stimulated echo method (*Lutti et al., 2014*; FOV = 256 × 192 × 192 mm$^3$, matrix = 64 × 64 × 48 mm$^3$, TE = 39.06, mixing time = 33.8 ms, TR = 500 ms, nominal $\alpha$ varying from 115° to 65° in steps of 5°, acquisition time 4 min 24 s). In total, the MPM protocol took approximately 20 min to acquire.

We did collect other imaging data as part of the Oxford BOLD study, including fMRI data (*Krishnan et al., 2021*). Notably, we also obtained a T1-weighted MPRAGE scan (magnetisation prepared low angle spoiled gradient echo, TR 1900 ms, TE 3.97 ms, flip angle 8°, field of view 208 × 256 × 256 mm) with 1-mm in-plane resolution and 1-mm slice thickness.

## Procedure

MPM data were collected at the end of the scanning session. The session also included two task fMRI scans, a resting state scan, and a diffusion-weighted scan; these data are not reported here. During the MPM scans, participants were given the option of either closing their eyes or watching an animated film; nearly all participants chose the film.

Participants also completed a comprehensive neuropsychological battery outside of the scanner, focusing on their linguistic and cognitive abilities (see *Krishnan et al., 2021* for further details). In brief, language ability was assessed using five tests, assessing aspects of expressive and receptive grammar, narrative, and vocabulary. Specifically, grammatical comprehension was assessed using the Test for Reception of Grammar – 2 or its electronic counterpart (TROG-E, *Bishop, 2005*). Expressive grammar was evaluated using the Recalling Sentences subtest of the Clinical Evaluation of Language Fundamentals – 4th Edition (CELF-4; *Semel et al., 2004*). Children's narrative production and comprehension were assessed using the Expression, Reception and Recall of Narrative Instrument (ERNNI; *Bishop, 2004*). Receptive and expressive vocabulary were assessed using the Receptive One-Word Picture Vocabulary Test – 4th Edition (ROWPVT-4; *Martin and Brownell, 2011b*) and Expressive One-Word Picture Vocabulary Test – 4th Edition (EOWPVT-4; *Martin and Brownell, 2011a*), respectively.

In addition to the language measures, children also completed the phonological decoding and sight word reading efficiency subtests of the Test Of Word Reading Efficiency (TOWRE; *Torgesen et al., 1999*); the block design, matrix reasoning, and coding subtests of the Wechsler Intelligence Scale for Children – 4th Edition (WISC-IV; *Wechsler, 2004*), the forward and backward digit span subtests, as well as the word lists subtest, from the Children's Memory Scale (CMS; *Cohen, 1997*), a nonword repetition test (*Norbury et al., 2016*), and the oromotor sequences subtest of the NEuroPSYchology (NEPSY) test battery (*Korkman et al., 1998*).

## Data pre-processing

Data were processed using the hMRI toolbox within SPM12 (*Balteau et al., 2018*; *Tabelow et al., 2019*). The default toolbox settings were used. This processing results in the MT saturation, R1 and R2* maps, which index different aspects of tissue microstructure. Briefly, R1 (1/T1) maps were estimated from the PD- and T1-weighted images using the process described in *Weiskopf et al., 2013*, extended by using correction for RF transmit field inhomogeneities and imperfect spoiling. Regression of the log signal from the signal decay over echoes across all three MPM contrasts was used to calculate a map of R2* (=1/T2*) (*Weiskopf et al., 2013*). RF transmit field maps were calculated from the 3D EPI acquisition and corrected for off-resonance effects as described in *Lutti et al., 2014*. The semi-quantitative MT saturation parameter (MTsat) calculated is relatively robust against differences in relaxation times and RF transmit and receive field inhomogeneities, and small residual higher dependencies are further corrected for within the toolbox.

Using quality assessment metrics obtained from the toolbox, we removed images where the SD R2* (a measure of image degradation) was greater than three times the interquartile range from the group mean. Data from nine children with DLD and two TD children were excluded on this basis. We also removed scans where interscan movement exceeded 2 mm. Data from a further four children with DLD, five TD children, and one child with HSL were excluded on this basis. We then conducted a visual inspection of the R1, MTsat, and R2* maps, and excluded data from a further five children with DLD, four children with HSL, and nine TD children where image artifacts were observed. We retained data from 56 TD children, 33 children with DLD, and 20 children with HSL. This equates to data retention of 77.78% in the TD population, 64.71% in the DLD population, and 80% in the HSL population.

Using pipelines implemented in the hMRI toolbox, MTsat maps from each participant were further segmented into grey and white matter probability maps. These grey and white matter maps were used to create a DARTEL template. Each participant's MTsat, R1, and R2* maps were registered to this DARTEL template and were then normalised to a standard MNI template. A tissue-weighted smoothing kernel of 6-mm full-width-at-half-maximum was applied using the voxel-based quantification approach (*Draganski et al., 2011*), which aims to preserve quantitative values for interpretation.

## Data analyses

We analysed group differences in MTsat, R1, and R2* maps using FSL's *randomise* tool using 5000 permutations. For assessing statistical differences across groups, we employed threshold-free cluster enhancement, setting p < 0.05 as our threshold. Data from regions of interest were extracted using *fslstats*, and further analyses were carried out using R (*R Development Core Team, 2020*).

For the behavioural data, we constructed factor scores for use in continuous analyses to minimise the number of comparisons in statistical tests. These factors were based on analysis of the whole cohort of children who contributed behavioural data to our study and therefore includes data from children who did not complete the MPM scans and children in whom we excluded MPM data as described above. The measures from the language and memory tests described above were entered into a pre-registered factor analysis to identify the best weighted combination of measures to give a language factor score, and a memory factor score. The approach we adopted to factor analysis was E-CFA (*Brown, 2006*), implemented in *lavaan* (*Rosseel, 2012*) in the R programming language (*R Development Core Team, 2020*). E-CFA is a hybrid exploratory–confirmatory approach to factor analysis where a model is specified with an 'anchor' measure or two anchor measures. As anchor measures, we used the list learning standard score from the CMS for the memory factor, and expressive vocabulary for the language factor. We planned to compare this two-factor model to a single-factor model accounting for language proficiency alone. However, our two preregistered models were not a good fit to the data. Consequently, as detailed in *Krishnan et al., 2021*, we accounted for

strong correlations between expressive and receptive vocabulary scores, as well as the two narrative production measures in modified models, and found the modified two-factor model to be a better fit to the data than the modified single-factor model. We consequently derived language and memory proficiency scores using this modified two-factor model.

## Acknowledgements

We thank all of our participants and their families, without whom this work would not be possible. We would also like to acknowledge the many individuals and organisations that helped us with recruitment (https://boldstudy.wordpress.com/acknowledgements/). We especially thank Professor Dorothy Bishop for her support, insight, and discussions throughout OxBOLD. We also thank members of the Wellcome Centre for Integrative Neuroimaging, especially the MRI team at the Oxford Centre for Human Brain Activity: Sebastian Rieger, Juliet Semple, Nicky Aikin, Nicola Filippini, Eniko Zsoldos, and Emily Hinson. We are grateful to Professor Fred Dick, as well as members of the Oxford Speech and Brain Group for helpful discussions and support. Funding: The Oxford Brain Organisation in Language Development or OxBOLD study was funded by the Medical Research Council MR/P024149/1 and supported by the NIHR Oxford Health Biomedical Research Centre. The Wellcome Centre for Integrative Neuroimaging is supported by core funding from the Wellcome Trust (203139/Z/16/Z).

## Additional information

### Funding

| Funder | Grant reference number | Author |
|---|---|---|
| UK Research and Innovation | MR/P024149/1 | Saloni Krishnan Kate E Watkins |
| Wellcome Trust | 203139/Z/16/Z | Saloni Krishnan Gabriel J Cler Harriet J Smith Hanna E Willis Salomi S Asaridou Máiréad P Healy Daniel Papp Kate E Watkins |

The funders had no role in study design, data collection, and interpretation, or the decision to submit the work for publication. For the purpose of Open Access, the authors have applied a CC BY public copyright license to any Author Accepted Manuscript version arising from this submission.

### Author contributions

Saloni Krishnan, Conceptualization, Data curation, Formal analysis, Funding acquisition, Writing - original draft, Project administration, Writing – review and editing; Gabriel J Cler, Data curation, Project administration, Writing – review and editing; Harriet J Smith, Hanna E Willis, Investigation, Project administration, Writing – review and editing; Salomi S Asaridou, Data curation, Supervision, Validation, Project administration, Writing – review and editing; Máiréad P Healy, Data curation, Validation, Project administration, Writing – review and editing; Daniel Papp, Conceptualization, Resources, Investigation, Writing – review and editing; Kate E Watkins, Conceptualization, Supervision, Funding acquisition, Investigation, Visualization, Writing - original draft, Project administration, Writing – review and editing

### Author ORCIDs

Saloni Krishnan http://orcid.org/0000-0002-6466-141X
Gabriel J Cler http://orcid.org/0000-0002-5279-7336
Harriet J Smith http://orcid.org/0000-0003-4314-6571
Hanna E Willis http://orcid.org/0000-0002-5479-6169
Máiréad P Healy http://orcid.org/0000-0002-4081-8275
Daniel Papp http://orcid.org/0000-0003-1481-1413

Kate E Watkins  http://orcid.org/0000-0002-2621-482X

### Ethics

This study was approved by the Medical Sciences Interdivisional Research Ethics Committee at the University of Oxford (R55835/RE002). Before enrolling participants in the study, we obtained written informed consent from parents/guardians, and written assent from children.

### Decision letter and Author response

Decision letter https://doi.org/10.7554/eLife.74242.sa1
Author response https://doi.org/10.7554/eLife.74242.sa2

---

## Additional files

### Supplementary files

• Supplementary file 1. Supplementary information.
 (a) Typically developing (TD) > developmental language disorder (DLD) differences in Magnetization Transfer saturation (MTsat) maps. Nonparametric randomisation analysis with threshold-free cluster enhancement was used to compare groups. A whole-brain corrected threshold of p < 0.05 was used. (b) TD > DLD conjoint differences in R1 and MTsat. (c) Differences in age and language scores between the selected and excluded children who were TD or had DLD.

• Transparent reporting form

### Data availability

The data that support the findings of this study are openly available on the OSF (https://doi.org/10.17605/OSF.IO/D93GQ). Statistical maps can also be viewed on Neurovault (https://neurovault.org/collections/DUGBDBPH/).

The following dataset was generated:

| Author(s) | Year | Dataset title | Dataset URL | Database and Identifier |
|---|---|---|---|---|
| Krishnan S, Watkins KE | 2021 | MPM | https://doi.org/10.17605/OSF.IO/D93GQ | Open Science Framework, 10.17605/OSF.IO/D93GQ |

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
