## [Editor Report]

The work establishes changes in striatal and cortical myelin as a neural basis for developmental language disorder (DLD). This is a significant advance in the understanding of this disorder that will stimulate further work.

---

## [Decision Letter]

**Decision letter after peer review:**

Thank you for submitting your article "Quantitative MRI reveals differences in striatal myelin in children with DLD" for consideration by *eLife*. Your article has been reviewed by 3 peer reviewers, and the evaluation has been overseen by a Reviewing Editor and Floris de Lange as the Senior Editor. The following individuals involved in review of your submission have agreed to reveal their identity: Faye Smith (Reviewer #2); Bob McMurray (Reviewer #3).

The Reviewing Editor has drafted joint editorial comments to help you prepare a revised submission.

Essential revisions:

Methodological

1. How critical is the use of the CATALISE definition to the results?

2. Why was the HSL group excluded from analysis?

3. The manuscript would benefit from specific analyses based on prespecified volumes of interest including whole striatum and IFG based on their previous hypothesis. The striatal prior is clear but the IFG prior might be justified further.

4. There are a lot of areas of signal change like occipital cortex not specified in advance that might be described in more detail.

Interpretational

1. Do the results inform debate about whether DLD at tail end of a distribution or a chunked disorder?

2. It is not essential but authors might speculate on how the changes in striatal myelination are related to the phenotype. The Oxford group has done a lot of work on the genetic of DLD: could any of that be related to gray-matter myelination? And they might also speculate on a mechanism at the neural level (note there is now an *eLife* structure for authors to fly kites more if they wish).

3. Referee 2 raises the important issue of causality and whether the striatal changes could be a consequence rather than a cause of DLD. The authors might acknowledge limitations of correlational approach and discuss.

*Reviewer #1 (Recommendations for the authors):*

1. I think the manuscript would benefit from specific analyses based on prespecified volumes of interest including striatum and IFG based on their previous hypothesis.

2. There are a lot of areas of signal change like occipital cortex not specified in advance that might be discussed in more detail.

3. It is not essential, but authors might specify how the changes in striatal myelination are related to the phenotype. The Oxford group has done a lot of work on the genetic of DLD could any of that be related to gray-matter myelination?

*Reviewer #2 (Recommendations for the authors):*

A clear rationale for why the HSL group was excluded from the whole-brain comparisons of microstructure would improve clarity. I found myself confused by the exclusion of that group for some analyses but not others. I think it would be interesting to see whether they sat somewhere between the DLD and TD groups or looked more similar to one or the other.

I think a little more detail about how the DLD group were characterised is important. The authors pointed to a previous paper, which is helpful for further details, but I think some clarity on what is meant by 'a history of language problems' is warranted in this paper too as it is key for understanding the nature of the clinical group.

Analyses that look at the correlation between brain microstructure and nonverbal ability, as well as with the language factor score would be valuable in demonstrating whether the differences are specifically related to language or are related to differences in cognition much more broadly.

Lines 375-377 were the ones that particularly jumped out to me as taking assumptions of causality between structure and function beyond what the data can support here.

In line 635, I think you want 'ROI's' or 'regions of interest' rather than 'ROIs of interest'.

In Supplementary Table 1 are the scores presented raw scores or standard scores? I think there might be a mix, looking at the numbers, but I think this needs to be clearer.

---

## [Author Response]

Essential revisions:Methodological1. How critical is the use of the CATALISE definition to the results?

Thank you for raising this important point. The CATALISE definition of Developmental Language Disorder rightly places less emphasis on the discrepancy between nonverbal and verbal skills. In particular, this opens up the diagnosis of DLD to those with nonverbal IQs between 70-85. Previously, such children would have been excluded from the diagnosis of DLD or SLI because their language scores were in line with (some might say “predicted by”) this level of nonverbal IQ. We assume the reviewer is asking whether the inclusion of children with low nonverbal IQ and low language ability has affected our results. To address this, we reanalysed our data excluding the children with IQs between 70-85 (there were 5 such children, all with DLD). Our analysis revealed the same pattern of group differences (Figure 1 —figure supplement 1 and Figure 2 —figure supplement 1) indicating that our results are not driven by the use of the CATALISE definition of DLD and inclusion of individuals with low IQ. We therefore think it is unnecessary to remove those with lower nonverbal IQs from our main analyses. We have added this to the paper, page 12.

*“The CATALISE definition of DLD rightly removes the criterion requiring a discrepancy between verbal and nonverbal skills, which in practice means broadening the phenotype to include children with low nonverbal IQ. To determine whether our group differences were in fact driven by the inclusion of children with low nonverbal IQs in addition to low language ability, we assessed differences only in children with nonverbal IQ scores >85. Group differences in MTsat and R1 maps were observed even when removing the five children with DLD who had nonverbal IQs between 70-85 (see Figure 1 —figure supplement 1 and Figure 2 —figure supplement 1), indicating these differences were not driven by the inclusion of children with low nonverbal IQs.”*

More broadly, the reviewers focused on whether nonverbal IQ could explain our findings. Our key finding was reduced MTsat and R1 values in the caudate nucleus bilaterally. We explored the unique contribution of language proficiency, verbal memory, and nonverbal IQ in these regions by conducting hierarchical regressions. We found that language proficiency was the strongest predictor of MTsat values in the caudate nucleus, again suggesting that these differences were specific to language ability and not indexing nonverbal IQ. We have now added these analyses to the paper, pages 13-14.

*“The DLD and TD groups differed from each other on language proficiency, memory, and nonverbal IQ (see Table 1). We consequently assessed whether variation in language proficiency, memory, or nonverbal IQ best explained the variation in R1 and MTsat values in the regions of the caudate nucleus where we observed TD > DLD group differences. We constructed stepwise regressions to evaluate the contribution of each of these factors, including children from the HSL group to maximise power (the pattern of analysis was the same even if we limited our analyses to children in the TD or DLD groups).*

*For MTsat values in the caudate nucleus, we found that a model with language proficiency alone (β=.014, p<.001) was the best fitting model, explaining 14.57% of the variance, F(1,106)=18.08, p<.001. For R1 values in the caudate nucleus, we again found that a model with language proficiency alone (β=.012, p<.001) was the best fitting model, explaining 13.19% of the variance, F(1,106)=16.11, p<.001.”*

2. Why was the HSL group excluded from analysis?

The HSL group was not included in any group comparison but these children were included in the continuous analyses. We explain our reasoning below. Also, we now define this group at the start of the Results section to avoid confusion (pages 6-7) and apologise that this was overlooked previously.

Children in the HSL, or history of speech and language difficulties group, were recruited on the basis of presenting with a history of speech and language difficulties but our testing revealed language abilities in the normal range. They were not necessarily neurotypical – many had received speech-language therapy or additional support for these challenges in the past. The group was highly heterogenous, with children having very different histories and profiles, which included dyslexia (and some with spelling difficulties), dyspraxia, short-term memory problems, or speech difficulties. We could not be certain that these individuals were simply “resolved” DLD cases. For these reasons, we did not include the children in our group analyses. However, our continuous analysis focused on individual variability in language proficiency, this allowed us to include these children in these analyses, thereby increasing our power. We hope the information we have added (pages 6-7) clarifies this decision in the manuscript.

*“As part of the Oxford BOLD study, we collected brain imaging data from 162 children between the ages of 10-15 years, as well as performing a detailed characterisation of their language and cognitive skills. All children in the study had a nonverbal IQ > 70. Children were categorised as DLD if they scored 1SD below the mean on two or more language tests, (N=57), and typically developing (N=77) if they scored +/- 1 SD of the mean on language tests. Behavioural testing in a further 28 revealed that they did not meet our criteria for DLD but presented with a history of speech and language problems (HSL). After quality control, we retained MPM data from 109 of these children, including 56 typically developing children, 33 children with DLD and 20 children with HSL. The children in the HSL group were excluded from comparisons of TD and DLD, but were included in continuous analyses which allow us to examine language variability. The three groups (TD, HSL, and DLD) did not differ in in terms of age (see Table 1).”*

3. The manuscript would benefit from specific analyses based on prespecified volumes of interest including whole striatum and IFG based on their previous hypothesis. The striatal prior is clear but the IFG prior might be justified further.

There were two ways we could have analysed our data, taking a whole-brain approach or an ROI approach. Since this data has not previously been obtained in children with DLD, we decided to take the whole-brain approach and use statistically robust thresholding to avoid false positives. Having done so, we believe our results are potentially informative for future studies which may use an ROI approach to focus on areas we find to be different between groups. To conduct ROI analyses on our data having done the whole-brain analysis would, we believe, be what is called “double-dipping”. We have therefore chosen not to restructure the paper on this basis. To satisfy the reviewers, we did pull out these ROI values and have posted this table on the OSF (https://osf.io/2ba57/); however, we feel that describing the data in this way does not add to our understanding.

With respect to justifying our focus on the IFG – The left inferior frontal gyrus is a key area for language processing and accordingly there are known differences in terms of structure and function in this area based on previous smaller scaler studies in developmentally language impaired children. We were particularly interested in the IFG for two reasons, first, as we found both structural and functional differences in our previous study of DLD, and second, as the inferior frontal gyrus is a major source of inputs to the caudate nucleus. We have now clarified this in the introduction.

Page 3 – *“In the available literature, grey matter changes have been noted in the left inferior frontal gyrus and the posterior superior temporal gyrus (Badcock et al., 2012; Gauger et al., 1997; Jancke et al., 2007; Lee et al., 2020; Plante, 1991; Preis et al., 1998). These are core regions for language processing, with language activation in these regions observed across the literature (Price, 2012) and across languages (Ayyash et al., 2021).”*

Page 4 – *“The striatum is structurally and functionally connected to regions associated with language production, with the head of the dorsolateral caudate nucleus receiving inputs from inferior frontal cortex, and the putamen receiving inputs from motor, premotor and supplementary motor cortex (Alexander et al., 1986; Jarbo and Verstynen, 2015; Lima et al., 2016).”*

4. There are a lot of areas of signal change like occipital cortex not specified in advance that might be described in more detail.

The whole brain analysis revealed group differences in the quantitative maps in the occipital cortex, which would not be predicted in a study of the neural correlates of DLD. We discuss these differences in the discussion, page 21, highlighting the fact that some of these changes may be linked to other associated differences in the groups (for example, co-occurring dyslexia). It is worth pointing out the these differences would not be revealed if we had chosen only to focus on a priori ROIs rather than use a whole-brain approach (see response above). These unexpected findings in particular require replication and could serve as priors for future studies of DLD.

*“The broad constellation of challenges faced by children with DLD might also explain some of the group differences in MTsat and R1 values. For example, many children with DLD have co-occurring motor challenges (Hill, 2001; Sanjeevan and Mainela-Arnold, 2019), which may be reflected in changes in the motor network. This might also account for differences in regions we did not have clear hypotheses about, for example, in the occipital lobe. Many children in our study faced reading challenges, and left occipital-temporal hypoactivation is associated with dyslexia (Paulesu et al., 2014; Richlan, 2012) and now genetic variation in literacy skills (see above, Eising et al., 2021). One way to disentangle the contributions of these different cognitive traits would be to conduct large-scale studies including those with multiple diagnoses and use a transdiagnostic approach to establish specific relationships between brain and behaviour (Siugzdaite et al., 2020).”*

Interpretational1. Do the results inform debate about whether DLD at tail end of a distribution or a chunked disorder?

We think this is an interesting point, but are not sure whether our data can fully speak to this. Broadly, we see that both continuous and categorical approaches are good fits to our data. We have elaborated on this point in the discussion.

Page 22 – *“There has been significant theoretical debate about whether DLD represents the tail end of a spectrum of language ability or if represents a group or several subgroups that are biologically distinguishable from those with poor language ability (Tomblin, 2011). Broadly speaking, there is now consensus in the field that these children represent the extreme end of a distribution (Bishop et al., 2016; Lancaster and Camarata, 2019; Tomblin and Zhang, 1999). We examined if our data was consistent with this idea by examining the distribution of our data. When plotted, the average values of striatal myelination for individual participants indicated overlap between the DLD and TD groups (see Figure 1). So not all children in the DLD had values outside of the range of values for the TD groups, as might be expected if we were observing a distinct sub-type. However, our continuous analysis does indicate that those who have the lowest values for striatal myelination have the lowest language factor scores. (see Figure 4). Our limited exploration of how these measures are distributed both within the DLD group and across the cohort is consistent with the idea that DLD is a spectrum disorder. Even so, further theoretically driven analyses that relate specific behaviour to these quantitative measures of brain microstructure might provide insight into whether there are distinct sub-types within DLD. Future studies should focus on understanding the full spectrum of DLD using large datasets, establishing if there are distinct brain-behaviour relationships within this broad category.”*

2. It is not essential but authors might speculate on how the changes in striatal myelination are related to the phenotype. The Oxford group has done a lot of work on the genetic of DLD: could any of that be related to gray-matter myelination? And they might also speculate on a mechanism at the neural level (note there is now an eLife structure for authors to fly kites more if they wish).

We now include the following paragraphs in the discussion.

Page 22 – *“The striatal differences we report are broadly consistent with our views on the dorsal striatum being involved in language learning (Krishnan et al., 2016), as well as with theories such as the procedural deficit hypothesis (Lum et al., 2014; Ullman and Pierpont, 2005), and the procedural deficit circuit hypothesis (Ullman et al., 2020).This may be linked to difficulties extracting and retaining the sequential regularities for language, or challenges in automatising the use of language rules. Our sample with DLD shows differences in linguistic tasks that draw upon these abilities, such as nonword repetition or oromotor sequencing, even though they were not selected on this basis. Orpella and colleagues (2021) recently demonstrated that the dorsal and ventral striatum were crucial for developing and acting upon predictions in a statistical learning task. The use of such sensitive functional tasks may be important to demonstrate the functional consequences of these changes. Understanding such learning mechanisms may also help us understand how to design better intervention for children with DLD, for example, this may be a reason to simplify and repeat rules during learning, providing greater opportunities for practice and habituation. We are now testing the role of repetition in enhancing language comprehension in DLD (Parker et al., 2021).”*

Page 19 – *“The genetic mechanisms that might drive these striatal changes are currently unclear. DLD is a highly heritable disorder but identification of genetic pathways has been difficult due to the broad phenotype and changes in diagnostic criteria (for a recent review see Mountford et al., 2022). In terms of monogenic disorders, the best known case is that of the KE family, the affected members of which had a rare point mutation in the gene FOXP2 causing verbal dyspraxia as well as the kinds of language learning difficulties seen in DLD. The mutation in FOXP2 was associated with reduced volumes of the caudate nuclei bilaterally in the KE family (Watkins et al., 2002), but it is unknown whether this was related to a quantitative difference in myelin content as seen here in DLD. FOXP2 is strongly expressed in the medium-spiny neurons of the striatum in humans and other species, and in many other neural and non-neural tissues (Lai et al., 2003). FOXP2 is a transcription factor affecting the function of many downstream target genes, including CNTNAP2 and FOXP1, which in turn have been linked to phenotypic features of DLD (Vernes et al., 2008; Lozano et al., 2015). Aside from identifying rare monogenic causes of DLD, other approaches have yielded common risk variants that explain variances in phenotypic features of DLD, including phonological skills (Newbury et al., 2009). A recent meta-analysis of DLD genome-wide association studies (GWAS) of reading and language traits with an N >30,000 found significant genome-wide association with word reading and distinct genetic variation of word and nonword reading, spelling and phoneme awareness that correlated with cortical surface area of the left superior temporal sulcus (Eising et al., 2021). Interestingly, in our study, the posterior part of the superior temporal sulcus in the left hemisphere showed significantly reduced MTsat and R1 values in children with DLD (see Figures 1, 2 and 3).”*

3. Referee 2 raises the important issue of causality and whether the striatal changes could be a consequence rather than a cause of DLD. The authors might acknowledge limitations of correlational approach and discuss.

We fully agree with this point, and indeed, already note this in the limitations section as highlighted below, page 23.

*“It is also unclear whether the neural changes we observe are the cause of DLD, or a consequence of having a language disorder. Longitudinal studies where children are followed over time are the best way to shed light on this issue.”*

In addition to the specific points raised above, we also addressed one of the reviewers points about some relevant literature we had not covered.

Page 22 – *“Our more global differences in myelination would also be consistent with theoretical views postulating that children with DLD show differences in the speed of processing, perhaps due to inefficiencies in information transfer through the brain (Kail, 1994; Miller et al., 2001). However, more recent theories point to more specific lexical processing differences in DLD. Children with DLD have been shown to have greater difficulty with lexical selection and inhibition of competitors, rather than the initial perceptual or phonological processing (Nation, 2014). Computational models suggest this could be due to increased levels of lexical delay (McMurray et al., 2010). Abnormalities in myelination could result in differing levels of inhibition for targets and competitors. Techniques like MEG would be better suited to test this hypothesis.”*